# Regulation of Error-Prone DNA Double-Strand Break Repair and Its Impact on Genome Evolution

**DOI:** 10.3390/cells9071657

**Published:** 2020-07-09

**Authors:** Terrence Hanscom, Mitch McVey

**Affiliations:** Department. of Biology, Tufts University, Medford, MA 02155, USA; terrence.hanscom@tufts.edu

**Keywords:** alt-EJ, polymerase theta, microhomology-mediated end joining, chromosome rearrangements, resection

## Abstract

Double-strand breaks are one of the most deleterious DNA lesions. Their repair via error-prone mechanisms can promote mutagenesis, loss of genetic information, and deregulation of the genome. These detrimental outcomes are significant drivers of human diseases, including many cancers. Mutagenic double-strand break repair also facilitates heritable genetic changes that drive organismal adaptation and evolution. In this review, we discuss the mechanisms of various error-prone DNA double-strand break repair processes and the cellular conditions that regulate them, with a focus on alternative end joining. We provide examples that illustrate how mutagenic double-strand break repair drives genome diversity and evolution. Finally, we discuss how error-prone break repair can be crucial to the induction and progression of diseases such as cancer.

## 1. Introduction

DNA double-strand breaks (DSBs) are highly dangerous lesions that arise with surprising frequency. By one estimate, up to ten DSBs per cell per day occur in humans [1]. To deal with these breaks, cells have evolved a variety of robust and conserved repair mechanisms. In many cases, these can faithfully restore the genetic information at the break site. However, each repair mechanism also brings with it a certain risk of mutagenesis, which varies depending on the specific type of repair and the context in which it occurs.

We are beginning to understand some of the factors that dictate repair pathway choice and how these may impact rates of mutagenesis. For example, certain DSB repair pathways are more active during particular cell cycle phases. Classical or canonical non-homologous end joining (c-NHEJ), which is typically accurate or results in small insertions and deletions, operates throughout interphase, while high-fidelity homologous recombination (HR) is active mainly during the S and G2 phases of the cell cycle [2]. Interestingly, a highly inaccurate form of DSB repair named alternative end joining (alt-EJ) also occurs mainly during S/G2. This cell cycle preference can be explained by the observation that both HR and alt-EJ require resection to form single-stranded DNA, a process that is upregulated in S/G2 [3]. However, the way that cells choose between HR and alt-EJ is less clear. Importantly, alt-EJ can operate in cells that are competent for both c-NHEJ and HR. Alt-EJ is associated with the generation of genomic diversity at sites prone to DSB formation and can facilitate cellular and organismal adaptation [4]. In addition, it is a driver of human disease and is associated with poor cancer prognosis [5,6,7]. Thus, a better understanding of the factors that promote the decision to use alt-EJ and other types of error-prone repair is needed.

In this review, we begin by cataloging the various types of error-prone DSB repair and examining their similarities and differences. Our main focus is alternative end joining, as it has been most closely tied to cellular transformation and genome evolution. We then delve into the various factors that drive cells to prefer one type of repair over another. We conclude by providing selected examples of how error-prone DNA repair promotes genetic diversity in different contexts and may act to drive genome evolutionary processes at both the cellular and organismal levels.

## 2. Types of Inaccurate Double-Strand Break Repair

Double-strand break repair mechanisms can be roughly grouped into two main categories: those that involve synapsis of broken ends and subsequent ligation, including cNHEJ, alt-EJ, and single-strand annealing (SSA), and those that involve templated DNA synthesis using a homologous sequence, including synthesis-dependent strand annealing (SDSA) and break-induced replication (BIR). While many of these repair mechanisms have the potential to faithfully restore the original DNA sequence, all can also result in error-prone repair. Some, like alt-EJ, are always mutagenic. Error-prone DSB repair can result in a variety of inaccurate repair events, including deletions, insertions (both templated and non-templated), deletions coupled with insertions, and chromosome rearrangements. In the next sections, we describe each of these repair mechanisms, highlighting their mutagenic potential, and discussing contexts which favor particular repair outcomes.

### 2.1. Classical Nonhomologous End-Joining (cNHEJ): A First Responder

cNHEJ is a rapid repair pathway that is commonly defined by the protein components that promote it. It is active throughout interphase and is the preferred DSB repair pathway in vertebrates [1,8,9]. cNHEJ begins with the binding of the Ku70/80 heterodimer to blunt DNA ends or those with less than 5 nucleotides (nt) of single-stranded DNA. This binding acts as an initial barrier to resection and end processing. Ku70/80 then recruits and activates the DNA-PKcs catalytic subunit, which together form the DNA-dependent protein kinase (DNA-PK). For DSB ends that are directly ligatable, the XRCC4-XLF complex forms a sleeve-like structure around the duplex, stabilizing it for ligation via DNA ligase 4 [1,8]. This outcome usually results in error-free repair. In cases where the ends cannot be directly joined, cNHEJ proceeds using a variety of end-processing factors (reviewed in [8]). Depending on the organism, these can include the Artemis nuclease, tyrosyl DNA phosphodiesterase (TDP), DNA polymerases mu and lambda, and the Mre11-Rad50-Xrs2/Nbs1 (MRX/MRN) complex [1].

cNHEJ is typically accurate or minimally mutagenic, causing 1-5 base pair (bp) deletions [10], 1-4 bp microhomology joins, and small insertions [11] (Figure 1). Sometimes, these minor genomic changes can be advantageous, such as during class switch recombination and V(D)J recombination that promotes antibody diversity in the vertebrate immune system. In other contexts, cNHEJ contributes to disease proliferation [8,12,13]. For example, in human cells, cNHEJ is responsible for a majority of chromosomal translocations resulting from ionizing radiation and nuclease-induced breaks [14].

### 2.2. Alternative End-Joining: A Quick and Dirty Fix

Alt-EJ was originally used to describe any end joining that occurs in the absence of c-NHEJ, but we now know that it also happens in cNHEJ-competent cells. Often, alt-EJ is used as a general term to describe a variety of pathways that vary based on mechanistic and genetic differences but have significant overlap. Alt-EJ is slower than cNHEJ [1] and tends to be used less frequently. For example, in studies of mammalian cells alt-EJ occurred at 10% the frequency of cNHEJ [15] and 10–20% the frequency of HR [16]. Alt-EJ is used to repair 0.5–1% of all DSBs in mammalian cells, but only 0.006% of all DSBs in yeast [8,17]. The lower rate of alt-EJ in yeast may be due to the absence of DNA polymerase theta, which has been shown to contribute to a majority of alt-EJ in organisms that have it (see below, and [10,18,19,20,21]). While the original view of alt-EJ was that it acts as a fail-safe mechanism to fix persistent chromosomal breaks, the prevailing model now is that it is a highly regulated, independent repair pathway that exists as a viable option even when both cNHEJ and HR are available. [8,16].

### 2.3. Alt-EJ Can be Defined by Repair Outcomes

One way that alt-EJ can be classified is according to the properties of the repair junctions that are created. The most common form of alt-EJ involves resection to create single-stranded DNA (ssDNA) at the break site and annealing of microhomologous sequences within the ssDNA. After annealing, non-homologous 3′ tails are removed and repair is completed by fill-in synthesis and ligation. This process, termed microhomology-mediated end joining (MMEJ), results in deletions of varying sizes (Figure 1) [22].

In budding yeast, MMEJ-induced deletions range from a few base pairs to several kilobases of DNA [23,24], while in mammalian cells MMEJ typically results in shorter deletions. In human osteosarcoma cells, MMEJ requires limited 5′–3′ resection of less than 20 nt [16], while in mouse embryonic fibroblasts MMEJ can occur with longer, 70 nt ssDNA overhangs [10]. Typically, mammalian MMEJ uses short microhomologies of 1–8 bp for rejoining. While longer microhomologies may exist further from the break site, they are infrequently used. Notably, this tendency for minimal end processing and the use of 1–4 bp microhomologies can make it difficult to determine a priori whether a given repair event is generated through cNHEJ or MMEJ [25].

Alt-EJ can also produce insertions at the break site, with or without an accompanying deletion. In many cases, the insertions appear to be templated from DNA flanking the break. To explain the etiology of these types of junctions, the synthesis-dependent microhomology-mediated end-joining (SD-MMEJ) model was proposed [26]. SD-MMEJ is similar to MMEJ but has an additional intermediate DNA synthesis step to create de novo microhomologies that are then used in an MMEJ-like process. One key aspect of SD-MMEJ is the transient formation of secondary DNA structures (loops or hairpins) that occur via the pairing of 1–4 bp direct or inverted repeats found near the break. These secondary structures serve as primers for limited DNA synthesis, likely by polymerase theta (see below). The nascent DNA then dissociates, either spontaneously or via helicase activity, and the newly synthesized DNA anneals with microhomologous sequences on the other side of the break. A similar form of SD-MMEJ, in which the initial synthesis of microhomology occurs following transient base-pairing across the break (trans SD-MMEJ), has also been described [27].

The signature of SD-MMEJ is the creation of a characteristic repeat motif, which distinguishes it from simple MMEJ (Figure 1). The repeat motif can be direct or inverted and includes both the secondarystructure-forming repeat and the microhomology used for annealing. While SD-MMEJ is most easily identified by the presence of templated insertions, the process can also produce simple deletions and apparent blunt-end joins. Insertion events with apparently untemplated nucleotides, which are commonly seen in alt-EJ, may represent multiple templated synthesis events, or they could be created through SD-MMEJ using an error-prone polymerase [26,27].

### 2.4. Alt-EJ Can be Defined by Genetic Requirements

Another way to classify alt-EJ is by the proteins that promote it. In budding yeast, MMEJ was originally characterized as a Ku-independent repair process that also occurs without the Rad52 protein, which distinguishes it from SSA [23]. Budding yeast MMEJ is promoted by Sae2 (CtIP) and Tel1 (ATM) and mutations in these genes confer an increase in blunt-ended ligation via cNHEJ [24,28]. In yeast, MMEJ also depends on Mre11, Rad50, and Rad1, and is modulated by Pol32, a non-essential subunit of polymerase delta, and Cdc9, the DNA ligase 1 catalytic subunit [23].

In vitro experiments using chicken DT40 B cells initially revealed roles for DNA ligases 1 and 3 and poly-ADP ribose polymerase (PARP) in an alternative end-joining mechanism that occurred in the absence of cNHEJ [29,30]. Many of the proteins required for alt-EJ were found to be the same as those required for the primary steps of HR, suggesting that these two repair processes share a common initial mechanism.

In 2010, a major role for DNA polymerase theta in alt-EJ was described in Drosophila [31]. Subsequently, polymerase theta was shown to be critical for alt-EJ repair of DSBs following replication fork collapse in *C. elegans* [20]. Since then, polymerase theta has been implicated in alt-EJ in Arabidopsis and many vertebrates [18,21,32,33], Because of the central role of polymerase theta in alt-EJ, many groups refer to alt-EJ as theta-mediated end joining (TMEJ) [10,34,35].

### 2.5. The Mechanism of TMEJ

Pol theta contains an A family polymerase domain at its C terminus which is essential for TMEJ. It facilitates end joining through a unique insertion loop in the polymerase domain that contacts the 3′ terminal phosphate of the primer DNA and promotes its extension using a short, microhomologous template. Pol theta can use as little as 1–2 bp of microhomology to prime synthesis of 1–25 nt of nascent DNA, with an average length of 3–6 nt [36]. Evidence from studies using mouse embryonic fibroblasts with Cas9-generated DSBs suggests pol theta can sample all microhomologous sequences for annealing within the 3′ terminal 25 nucleotides of a pair of ends [10]. Iterative rounds of the annealing, synthesis, and dissociation steps can lead to insertions templated from regions around the DSB [20,32,37].

Pol theta also contains an ATPase-dependent SF2 helicase domain at its N terminus. This domain has been shown to unwind short pieces of duplex DNA [38] and remove replication protein A (RPA) from ssDNA [39]. The helicase domain can also anneal complementary ssDNA ends [19], consistent with its role in promoting TMEJ. Further investigation is needed to determine if the helicase domain can also unwind newly synthesized microhomology intermediates in SD-MMEJ prior to the final annealing step.

Recently, the importance of the pol theta helicase domain was demonstrated in vivo. In Drosophila, pol theta ATPase-dead and null mutants are equally sensitive to interstrand crosslinking agents. In addition, ATPase-dead mutants produce significantly fewer insertions during alt-EJ and use shorter microhomologies at repair junctions [40]. In human cells, pol theta ATPase activity is required for survival in an HR deficient background [41]. Thus, the helicase domain is vitally important for TMEJ, a property that could be exploited in HR-deficient cancers [42].

Pol theta plays an important role in the repair of one-ended DSBs and breaks with damaged bases or protein adducts that cannot be readily ligated [43]. It promotes mutagenesis via deletion-prone repair and by its high nucleotide misincorporation rate [21]. In the absence of pol theta, another type of cNHEJ-independent end-joining mechanism still operates, producing very large deletions at sites of DSBs [4,10,20,31] (Figure 1). In the *C. elegans* and Drosophila germlines, loss of polymerase theta is associated with deletions of multiple kilobases of DNA, suggesting that the mutational cost of TMEJ is small compared to the genomic catastrophe that results in its absence [4,10,34].

### 2.6. When Homologous Recombination “Goes off the Rails”

HR repairs double-strand breaks by copying missing or damaged information from a homologous template, such as a sister chromatid, and typically ensures faithful reproduction of the original sequence. HR, like alt-EJ, requires an initial resection step that is carried out in metazoans by the MRN complex and phosphorylated CtIP. Continued extensive resection, catalyzed by the BLM helicase and DNA2 endonuclease or the EXO1 exonuclease, differentiates HR from alt-EJ. The resulting 3′ ssDNA ends are coated with RPA, which is then exchanged for the RAD51 recombinase. The DNA-RAD51 nucleoprotein filament initiates the search for a suitable template through the formation of a displacement loop (D-loop). The paired 3′ single-stranded DNA is then extended via polymerase activity [16,44]. In most instances, the D-loop dissociates after synthesis of several hundred nucleotides and the nascent strand pairs with single-stranded DNA on the opposite side of the break in a process called synthesis-dependent strand annealing (SDSA, Figure 2) [44]. Alternatively, the second end of the DSB can be captured prior to the dissociation of the D-loop, resulting in a double Holliday junction structure that must be cleaved to finish the repair. This mechanism is commonly referred to as the double-strand break repair (DSBR) model [45].

Although HR is typically error-free, there are contexts where it can result in genome instability. For example, invasion of the RAD51-coated ssDNA into a similar but not identical sequence can result in non-allelic or ectopic homologous recombination. Non-allelic HR (NAHR) can lead to copy number variants of short sequence motifs or entire genes (Figure 2). In a study investigating 17 human cancer genomes, NAHR was estimated to be responsible for 22% of all structural variations [12,46]. Nearly 45% of the human genome is made up of repetitive elements that can contribute to ectopic recombination during DSB repair, leading to chromosomal rearrangements associated with human disease [47,48]. For example, Alu repeats are ~300 bp highly homologous sequences that represent ~11% of the overall DNA in the human genome. Strikingly, 0.3% of all new human genetic diseases are caused by ectopic recombination between Alu repeats [49,50]. As one example, 42% of the intronic sequences of BRCA1 are made up of Alu repeats that range in size from 0.5 kb–23.8 kb, and erroneous recombination between them is one of the most frequent causes of mutation in BRCA1 deficient cancers [49,50].

### 2.7. Break-Induced Replication

BIR is a highly mutagenic HR pathway that was first described in yeast at one-ended double-strand breaks but has since been shown to operate in human cells at persistently collapsed replication forks and during the repair of eroded telomeres in both yeast and human cells [51,52]. BIR requires the resection proteins involved in SDSA and is Rad51-and Rad52-dependent in budding yeast but is sometimes Rad51-independent in mammalian cells [51,53]. DNA synthesis in BIR is conservative and is mainly carried out by DNA polymerase delta, with small contributions from polymerases epsilon and alpha [51]. BIR differs from SDSA in that BIR generates extremely long DNA synthesis tracts of up to 10 kb in mammalian cells and >100 kb in yeast [54] (Figure 2). While one-ended DSBs are a preferred substrate for BIR, limited homology of less than 150 bp at the non-invading end of a two-ended DSB causes inefficient second end capture and can also bias repair toward BIR [55,56].

BIR can cause extensive mutagenicity in the form of loss of heterozygosity, translocations, dicentric chromosomes, and copy number variations. In addition, it is 1000-fold more mutagenic than replication in yeast due to the ssDNA that accumulates due to asynchronous synthesis of the leading and lagging strands. This ssDNA is highly susceptible to base damage and is a prime contributor to BIR mutagenicity [57]. Lastly, BIR at damaged replication forks in mammals has been reported to result in genomic duplications seen in many cancers [55].

At the extreme end of mutagenic HR is microhomology-mediated break-induced replication (MM-BIR). During MM-BIR, ssDNA created during the early stages of BIR dissociates from the initial template and temporarily anneals with microhomologous sequences at other locations in the genome (Figure 2). MM-BIR initiates with low processivity synthesis and can involve multiple template switches, generating complex rearrangements that are occasionally seen in mammalian cancers and other human diseases. These template switches can involve microhomologies of only 2–5 bp [58]. This use of short microhomologies distinguishes the mechanism of MM-BIR from the initial strand invasion steps of traditional BIR [59]. Overall, the insights garnered from recent studies of MM-BIR demonstrate that rampant template switching can be a potent driver of genome instability and evolution.

### 2.8. Single-Strand Annealing: Not Just Extreme MMEJ

SSA is a conserved DSB repair mechanism that was initially classified as a form of HR because both repair processes require Rad52 in budding yeast. Like HR, SSA requires resection and the formation of 3′ ssDNA ends. However, instead of RAD51-mediated strand invasion, SSA repair proceeds by annealing of homologous sequences, clipping of 3′ single-stranded tails, and ligation [60]. Like MMEJ, SSA results in deletions of the region between the homologous sequences that initially anneal to each other. However, SSA differs from MMEJ in that it uses longer microhomologies than MMEJ (>15 bp) and does not require DNA pol theta. Extensive resection may favor SSA over alt-EJ due to an increased probability of exposing longer microhomologies, especially in the context of vertebrate genomes with abundant transposons and long repeats [60].

## 3. Factors That Affect Repair Pathway Choice

Many factors are implicated in the DSB repair pathway decision, including the complexity and number of DSBs, stage of the cell cycle, genome size, genome organization, and the presence of repetitive DNA. In addition, the availability of repair proteins and their post-translational modification status can influence what repair mechanisms are used. In the next sections, we highlight some of the most important factors for DSB repair pathway choice, with an emphasis on those that drive mutagenesis and genome diversity (Figure 3).

### 3.1. Cell Cycle Position

In many organisms, the decision between HR and cNHEJ is tightly regulated by cell cycle-dependent phosphorylation of DNA repair proteins [9]. In budding yeast, the cyclin-dependent kinases CDK1 and CDK2 are upregulated during S and G2, when they downregulate cNHEJ by opposing Ku and DNA ligase 4 binding and phosphorylate Sae2 to promote resection [61]. In human cells, CDKs also phosphorylate the Sae2 ortholog, CtIP, at multiple residues to promote resection [62].

While the preference for HR or cNHEJ varies widely between different organisms, the mechanisms that determine preference are similar. In both yeast and mammals, Mre11 responds to DSBs throughout the cell cycle and is involved in both HR and cNHEJ [61]. In yeast during G1, the Ku heterodimer promotes cNHEJ and limits the endonuclease activity of Mre11 to inhibit resection. During G2, HR is the dominant DSB repair pathway and yeast will only perform cNHEJ if HR is perturbed [63]. In contrast, the dominant DSB repair pathway in human cells throughout interphase is cNHEJ, with HR and alt-EJ serving important roles during S and G2. For example, following exposure to ionizing radiation, approximately 80% of IR-induced DSBs in human fibroblasts are repaired by end joining. While most repair is likely carried out by cNHEJ, it is likely that some of the breaks are repaired by alt-EJ, as repair junctions with small insertions or deletions are characteristic of both pathways [61,64,65]. Interestingly, clusters of IR-induced DSBs require alt-EJ for their repair, suggesting that some radiation damage may inhibit repair by cNHEJ [65]. During G1, HR is repressed and cNHEJ performs an even greater proportion of DSB repair. The reason why mammals rely so heavily on cNHEJ over HR in comparison to yeast may have to do with the size and composition of the genome and the potential for deleterious rearrangements via HR in the repeat-rich mammalian genome.

Until recently it was thought that both HR and cNHEJ were inhibited during mitosis due to partial inactivation of the DNA damage response (DDR), which is a network of proteins that sense, signal, and repair DNA lesions [9]. However, recent studies have shown that DSB repair enzymes from both HR and cNHEJ respond to ionizing radiation-induced DSBs in metaphase and that the completion of repair is delayed until G1 [66]. Proteins such as the MRN complex, the complete cNHEJ complex, as well as early HR factors were found to be enriched at mitotic DSB sites while other proteins involved in DSB repair were not recruited [67,68]. Currently, it is thought that early repair factors like MRN may keep DNA ends tethered together until G1, when the repair can be completed [66]. Tethering DSB ends together is important for proper chromosome segregation during cell division. Untethered fragments may become micronuclei that when reincorporated can induce chromothripsis, which is a devastating series of genomic rearrangements frequently associated with cancer (expanded upon below).

### 3.2. Resection

End resection involves a multitude of proteins and is highly regulated. In budding yeast, the MRX complex is thought to initially direct repair towards cNHEJ. Following activation of the endonuclease activity of Sae2 via CDK phosphorylation, the MRX-Sae2 complex cleaves the 5′ strand internally. Similarly, in mammalian cells, CDK phosphorylation of threonine 847 in CtIP promotes resection, while phosphorylation of serine 327 promotes CtIP interaction with BRCA1, initiating the commitment to HR [64,65]. These activities also remove barriers to resection, including protein adducts such as stalled topoisomerases and the Spo11 meiotic recombination protein, along with DNA secondary structures and lesions [62,65]. In yeast, resection is further potentiated by the recruitment of Tel1 by the MRX complex and Mec1 by RPA-coated ssDNA. RPA-ssDNA promotes long-range resection by the Exo1 exonuclease and the Sgs1 helicase/Dna2 endonuclease.

In metazoans, additional regulators of resection and repair pathway choice exist. One important protein is BRCA1, which promotes resection during G2 and whose loss impairs RAD51-mediated recombination. 53BP1 promotes cNHEJ throughout interphase and is one of the main barriers to resection. At DSBs in G1, 53BP1 is phosphorylated by ATM, stimulating interaction with its effector, RIF1, to inhibit BRCA1 mediated resection [69]. During S/G2, BRCA1 blocks the interaction of RIF1 and 53BP1, allowing resection to occur. Mutations in BRCA1 result in resection defects that can be rescued by loss of 53BP1 [62]. Therefore, BRCA1 is not essential for resection but its role is rather to surmount the energetic barrier to resection by antagonizing 53BP1 [9,62,70]. Recently, 53BP1 and RIF1 were also shown to recruit the shieldin complex, composed of SHLD1, SHLD2, SHLD3, and REV7, to DSBs in G1 [71]. Thus, the combined action of 53BP1-RIF1-shieldin serves to block resection in G1, ensuring that it occurs only when a sister chromatid template is present.

End resection is one of the main determinants of DSB repair pathway choice (Figure 3). The initiation of end resection is inhibitory for cNHEJ. In contrast, small amounts of resection expose microhomologies close to the break site and provide opportunities for alt-EJ. Continued resection that creates long stretches of single-stranded DNA is necessary for HR and SSA [72]. After long-range resection occurs, if a homologous template is not available for HR repair, then SSA becomes more likely.

Although resection occurs mainly in S and G2, limited resection has also been reported during G1 in human cells. This limited resection appears to be mediated by MRN and phosphorylated CtIP and is enhanced by the 53BP1 protein [73]. Rather than the CDKs, the Polo-like-kinase 3 is responsible for phosphorylating resection machinery in G1 [74]. This suppresses cNHEJ and promotes alt-EJ [73]. While alt-EJ that occurs in G1 is slower and more mutagenic than cNHEJ, it may be necessary for the repair of complex breaks that occur in G1 [75].

DNA packaging can facilitate or inhibit DSB repair by altering enzyme accessibility to damage. Therefore, a highly controlled chromatin response to a DSB is important to promote repair fidelity [76]. Often, chromatin remodeling enzymes are needed for the movement of nucleosomes to allow for efficient resection. One example is the budding yeast Fun30 chromatin remodeler that is an important mediator of both Sgs1/Dna2- and Exo1-dependent resection [77]. The human ortholog SMARCAD1 performs a similar function and its perturbation impairs end resection, HR, and sensitizes cells to PARP inhibitors [78].

### 3.3. RPA

Replication protein A is a heterotrimeric ssDNA binding protein involved in all ssDNA transactions in the nucleus. RPA has both low- and high-affinity binding modes and undergoes constant local dissociation and exchange via these modes [79]. It is required for replication, recombination, preventing spontaneous annealing of single-stranded DNA, and protecting ssDNA from degradation by nucleases. RPA prevents mutagenesis by reducing the formation of hairpins and other secondary structures at repeated sequences, including trinucleotide repeats and palindromes. These structures have the potential to create dicentric chromosomes and tandem duplications that are often seen in cancer genomes [80].

RPA promotes the initiation of recombination-based repair pathways through several different mechanisms. In budding yeast, RPA bound to ssDNA enhances the endonuclease activity of MRX-Sae2 to promote HR in S/G2 [81]. RPA is also required for Sgs1/Dna2-mediated long-range resection. Following Sgs1 unwinding of DNA at a break, RPA binds to and protects the 3′ strand and directs Dna2 to cleave the 5′ strand. While RPA is not required for Exo1 recruitment to resected DSBs, the Exo1-mediated resection pathway is dysfunctional without RPA [80,82].

In contrast, RPA inhibits MMEJ, which relies upon annealing of ssDNA. RPA likely also inhibits SD-MMEJ, although this has yet to be formally demonstrated. Overexpression of yeast Rfa1, the largest RPA subunit, suppresses MMEJ by inhibiting spontaneous annealing. In contrast, the expression of a mutated form of Rfa1 that is defective in DNA binding confers a 350-fold increase in MMEJ [83]. These properties of RPA are likely relevant to cancer cells where replication stress may lead to abundant ssDNA. Physiological levels of RPA may be insufficient in this context, leading to increased alt-EJ [84,85].

### 3.4. Microhomology

During alt-EJ, the extent of microhomologous sequences flanking the break site and the microhomologies themselves can affect the efficiency of repair. One important consideration is the size of the microhomology. In budding yeast, the frequency of MMEJ is proportional to the length and melting temperature of the microhomologies, as the frequency of MMEJ increases 10-fold for every additional base pair of microhomology between 12–17 bp [86]. Similarly, in fission yeast, even a single heterologous base at the 3′ end reduces MMEJ efficiency two-fold [17,87]. Microhomologies used in MMEJ in budding yeast are longer than in metazoans, requiring at least 8 complementary base pairs in one study [88]. In contrast, the microhomologies used in metazoan alt-EJ can be as short as 1 bp, likely due to the annealing action of polymerase theta.

A second factor relevant to alt-EJ is the proximity of the microhomologies to the break. In one study using mammalian cells, the efficiency of TMEJ was high when 2–4 bp microhomologies were directly adjacent to a DSB but dropped significantly when the distance from the break was increased to 10–30 bp. Increasing the length of microhomology to 6 bp rescued this distance-dependent decrease, illustrating that successful alt-EJ depends on a balance between the size of the microhomology and its location relative to the break [37,86].

The DNA context adjacent to the DSB also affects the proficiency of alt-EJ. In a study using mouse embryonic fibroblasts (MEFs), GC-rich flanking sequences produced more stable microhomologies and favored MMEJ while AT-rich flanking sequences were utilized 75% less often during MMEJ. This study also found that doubling an AT-rich microhomology from 3 bp to 6 bp increased the efficiency of MMEJ, highlighting a balance between microhomology length and AT/GC content. For alt-EJ outcomes that involved insertions, flanking sequences with 80% AT content produced more insertions than sequences with 80% GC content [37]. This bias may be due to the propensity of AT-rich microhomologies to dissociate following the initial MMEJ repair attempt, leading to SD-MMEJ and insertion formation.

As previously described, a defining feature of SD-MMEJ is the formation of secondary structures (due to annealing of short repeats) that can be used for the initial DNA synthesis step, creating de novo microhomologies. Three parameters related to these repeats are important for SD-MMEJ outcomes: repeat length, distance from the repeat to the break, and distance between the repeats. As might be expected, the most frequently used repeats are those that are closest to the break site. As structure-forming repeats approach a distance of 30 bp from the break site, their utilization decreases to near zero [27]. Similarly, the overall efficiency of SD-MMEJ decreases as the distance between the repeats decreases. When the distance between a commonly used GGCC repeat increases from 5 to 15 base pairs, its usage during SD-MMEJ declines in a stepwise manner (our unpublished data).

Perhaps unexpected is the finding that increased length of structure-forming repeats does not correlate with higher efficiency of SD-MMEJ. Studies in Drosophila using sequence variants flanking I-*Sce*I-induced breaks have shown that the preferred length of these repeats is only 2–4 bp, while repeats of >8 bp are utilized with low efficiency [27]. This suggests that the formation and dissociation of secondary structures is a highly dynamic (and perhaps regulated) process during SD-MMEJ.

Homology length also influences outcomes of HR repair pathways (Figure 3). Both SDSA and BIR are regulated by the amount of homology between the two DSB ends and the donor template. In yeast, if homology at the second end is ≤150 bp, second end capture is inefficient and repair shifts from gene conversion to BIR [56], but if significant homology is present SDSA outcompetes BIR. Shorter homologies between donor and recipient result in decreased viability and rates of repair [56]. Any heterology between the two sequences also decreases their recombination. For example, 3 nt of nonhomologous 3′ DNA is enough to significantly reduce recombination and promote heteroduplex rejection and unwinding of mismatched DNA [89].

## 4. How Error-Prone Repair Relates to Genome Evolution

Evolution occurs via mutation of DNA and natural selection for mutations that increase reproductive fitness. Error-prone DNA repair can, therefore, serve as a driver of genome evolution. To ensure high-fidelity repair of DSBs during germ cell formation, break repair during meiosis is preferentially completed by HR. While cNHEJ is the major DSB repair pathway in somatic tissues in metazoans, it is suppressed during meiosis in Drosophila and *C. elegans* [90,91,92]. Still, mutational signatures common to different types of error-prone DSB repair have been found in the germline of organisms from roundworms to humans, suggesting these mechanisms can contribute to evolution.

The extent to which different organisms display a bias for one type of repair over another contributes to evolutionary diversification. As one example, a study investigating genome size change in *Arabidopsis thaliana* and barley determined the Arabidopsis genome has been shrinking over time due to frequent genome deletions, while the barley genome has generally remained constant in size, with a greater incidence of genome insertions. These trends could result from differential reliance on error-prone DSB repair pathways, different tolerances to genome loss and gain, and different genome structural features that may influence these changes [93]. To elucidate which of these factors are most important in various genome contexts, further investigation is needed. Still, some published studies provide initial clues. We describe these in the next section and point out the remaining gaps in our understanding.

### 4.1. Alt-EJ Promotes Genome Evolution

There is substantial evidence that alt-EJ can act in the germline and thereby plays an important role in genome evolution. Whole-genome sequencing studies of *C. elegans* passaged over multiple generations have demonstrated that genome changes with signatures of polymerase theta accumulate over time. These include deletions with 1 or more nucleotides of microhomology and small templated insertions [4]. These alt-EJ genome scars are also observed in wild *C. elegans*. Notably, > 95% of error-prone DSB repair following transposon mobilization or Cas9 breakage in the *C. elegans* germline is pol theta dependent and mutations in HR or cNHEJ do not affect the frequency or pattern of these alterations [4].

Several organisms, including larvacean species like *O. dioica*, have completely lost the ability to repair DSBs by cNHEJ. In these cases, a form of alt-EJ that produces deletions with 4 bp microhomologies and templated insertions appears to be a prominent repair option [94]. Unlike in *C. elegans*, it is less clear how the genome of *O. dioica* has been influenced by this reliance on more error-prone end joining. It has been suggested that a preference for alt-EJ may be transformative for genome organization, as more genome rearrangements are observed in the absence of the cNHEJ protein Ku [2,94].

African trypanosomes, a type of parasitic flagellate protozoa, are also reported to lack cNHEJ machinery and rely on MMEJ for end-joining repair. A comprehensive study of DSB repair junctions in trypanosomes reveals microhomologies between 5–20 bp with a mean deletion size of 54 bp and a tolerance for mismatches within long stretches of microhomology [95]. Consistent with MMEJ observed in other organisms, proximity to the break, number and proportion of matched bases, and GC content all influence MMEJ in this organism. It is unknown whether trypanosomes have an enzyme similar to polymerase theta, but indications from other protozoan parasites suggest that it may and the DSB repair signatures are consistent with TMEJ [96]. Interestingly, a type of microhomology-mediated HR that leads to genomic rearrangements was also reported in trypanosomes [95].

### 4.2. Genome Compaction

The DSB repair processes that drive genome compaction, while incompletely understood, likely involve a variety of error-prone mechanisms, including cNHEJ, alt-EJ, and non-allelic homologous recombination. One process that is associated with a reduction in genome size is intron loss. A study of *Saccharomycetaceae* showed that intron loss almost always results in the complete loss of an intron without deletion of flanking coding sequence, consistent with HR replacement of the intron-containing genomic locus with a reverse transcribed copy of the corresponding mature mRNA [97,98]. Less than 3% of intron loss events in these yeasts are associated with loss or gain of < 3 codons, suggesting that cNHEJ and MMEJ play a minor role in this process. In Drosophila, intron loss has also been associated with alt-EJ involving microhomology annealing within a gene [99].

Transposable elements are mobile sequences whose movement can restructure genomes. Many transposons contain direct long terminal repeats (LTRs) that can participate in SSA, resulting in the deletion of most of the transposon. Such events leave behind a solo LTR and have been well documented in *Arabidopsis thaliana* [93]. Similarly, the genome of *Oryza brachyanth*, a type of wild grass in the rice genus, has contracted greatly during evolution via a combination of SSA involving transposon LTRs and also through cNHEJ [100]. In mammals, long interspersed nuclear elements (LINEs) are a type of transposable element that makes up approximately 17% of the genome. DSBs created in LINEs in murine cells are repaired via cNHEJ approximately 95% of the time, although non-allelic HR between homeologous LINE sequences occurs with appreciable frequency [101]. Comparison of the human and chimpanzee genomes suggests that both non-allelic HR and cNHEJ involving LINEs has resulted in deletions comprising about 450 kb during human evolution [102].

### 4.3. Genome Expansion

Genomes can also increase in size through several mechanisms, some of which are associated with inaccurate DSB repair. A study of the rice genome suggested that inaccurate DSB repair likely causes insertions between 10–1000 bp. These can involve tandem or partially tandem duplications, short duplications from ectopic donors, and non-tandem duplications with insertions corresponding to DNA sequences located within 200 bp of a break, often from multiple locations. The investigators hypothesized that these types of duplications resulted from the formation of DSBs by synchronous, asymmetrical nicks during base excision repair, followed by end joining [103].

Genome expansion can also occur through the endosymbiotic gene transfer of organelle genes to the nuclear genome. Foreign genetic elements in the nucleus are most often degraded upon transfer but are occasionally integrated downstream of a strong nuclear promoter via error-prone double-strand break repair mechanisms [104]. In *Arabidopsis*, up to 18% of nuclear genes appear to be derived from chloroplasts and their ancestor organelle [105]. An even greater percentage of yeast genes appear to have been transferred from mitochondria [106]. Similarly, many insertions in the Drosophila genome have occurred via end-joining repair with mitochondrial DNA [99]. In humans, fragments of DNA that result from oxidative damage in the mitochondria can be inserted into the nuclear genome via cNHEJ. The presence of microhomology and templated additions up to 11 bp suggest that alt-EJ also mediates these insertion events [107,108].

Most genome evolution occurs via a combination of both expansion and contraction events, the most well-known example of which can be found in *Saccharomyces cerevisiae*. Its genome is believed to have evolved via a whole-genome duplication event followed by a loss of nearly 90% of the duplicated genes via deletion-prone DSB repair [109]. The remaining duplicated genes evolved novel but often related functions as paralogs. A second example of rapid genome evolution was demonstrated using a highly novel genome restructuring method, in which yeast and Arabidopsis genomes were evolved following induction of multiple DSBs by a heat-inducible endonuclease. In this study, the evolved strains were found to harbor several rearrangements, copy number variations, and translocations involving retrotransposons that likely occurred through both cNHEJ and alt-EJ [110].

### 4.4. Error-Prone Repair Contributes to Genome Evolution in Cancer

Cancer cells experience severe replication stress that can lead to DSBs; this replication stress is exacerbated by exposure to chemotherapeutic agents that damage DNA. To deal with this stress, cancer cells often increase their reliance on error-prone DSB repair pathways, which contributes to mutagenesis [111,112]. Indeed, the mutational signatures observed in many different types of cancers often provide clues to the types of repair favored in each context [113]. Several groups have identified a “BRCAness” signature that is associated with HR-deficient cancers, commonly referred to as Signature 3 in the Catalog of Somatic Mutations in Cancer (COSMIC). Signature 3 is characterized by an elevated number of base substitutions, deletions, and insertions > 3 bp with microhomologies located at the suspected breakpoint junctions, characteristic of alt-EJ repair [114].

Mutations in the BRCA1 or BRCA2 genes result in a defect in HR repair, which causes cancer cells with these mutations to become “addicted” to error-prone alt-EJ pathways for their survival. Indeed, perturbation of alt-EJ repair in HR-deficient backgrounds can result in synthetic lethality. One illustration of this is the sensitization of BRCA1- or BRCA2-deficient cancers to PARP inhibitors. Because PARP1 promotes alt-EJ, its impairment in an HR-defective background leaves cells with no repair options. Furthermore, many PARP inhibitors trap PARP1 at ssDNA breaks, which obstructs DNA replication and leads to DSBs that would normally be repaired by HR [5]. In addition to cancer cells lacking BRCA1/2, many others with HR deficiencies are reliant on alt-EJ for their growth, including those with mutations in RAD51, CHD1, PALB2, and FANCD2 (Table 1).

A second example of alt-EJ addiction is the synthetic lethality observed when polymerase theta is removed or inhibited in HR-deficient backgrounds [115]. Pol theta promotes DSB repair with the junctions characteristic of those found in HR-deficient cells. Many HR deficient cancers overexpress pol theta, relying upon it for survival, and higher levels of pol theta correlate with increased cancer severity and poorer outcomes [7,116]. For these reasons, pol theta shows great promise as a cancer therapeutic target [42].

### 4.5. Genome Rearrangements

Genome rearrangements, including chromosome translocations, copy number variations, and telomere fusions, are frequently created through aberrant DSB repair. The repair pathways involved in their genesis differ widely between organisms. For example, translocations in budding yeast typically involve 2–20 bp microhomologies and are elevated in cNHEJ-deficient cells [86]. Mutations in the cNHEJ antagonist Sae2 also decrease translocations [133]. These findings suggest that in yeast, cNHEJ protects against translocations and a polymerase theta-independent form of alt-EJ mediates translocations.

In metazoans such as *C. elegans* that possess pol theta, TMEJ has been shown to generate most chromosome rearrangements in the germline [4]. Similarly, alt-EJ has a dominant role in translocation formation in mice, to a far greater degree than cNHEJ [86,111,134,135]. Loss of Ku or DNA ligase 4 causes a 3-fold increase in translocation frequency in MEFs [10]. Similarly, MEFs deficient in polymerase theta have reduced translocations in non-telomeric sequences and the remaining translocations display less use of microhomology [18,39]. In contrast, translocations in human cells are largely caused by cNHEJ. Translocation frequency decreases in DNA ligase 4 mutants [14,136]. While 31% of balanced translocations and inversions in human cells use microhomology, alt-EJ seems to be less important for their formation [12]. The reasons behind these differences in mice and humans are not clear.

Genome rearrangements are prevalent in cancer genomes. Distributions of rearrangements are highly variable and depending on the type of cancer, can be relatively even or clustered in hot spots within the genome. Common rearrangements seen in many cancers involve tandem duplications (TDs) between 3 kb and 1 Mb. These TDs often have microhomology at their ends, suggesting that they may result from alt-EJ repair [137]. These rearrangements cause structural changes in DNA that alter proteins, drive aberrant transcriptional patterns, and modify the chromatin environment. While many rearrangements are detrimental to cells, some can confer a fitness advantage, driving cancer progression [137].

Translocations can also occur at degraded telomere ends. Over time, shortened telomeres can cause genome instability via end-to-end fusions and large-scale genomic rearrangements, predisposing to cancer. In human cells, telomere fusions can be mediated through both DNA ligase 3 (alt-EJ) and DNA ligase 4 (cNHEJ), but cells that escape rearrangement crisis preferentially utilize cNHEJ [138].

### 4.6. Chromothripsis

Chromothripsis is a highly mutagenic process whereby the genome acquires dozens to thousands of simultaneous rearrangements. It was first observed in a chronic lymphocytic leukemia that contained 42 genomic rearrangements, all within the long arm of chromosome 4 [139]. Cancer cells frequently contain massive rearrangements that are consistent with a chromothripsis mechanism; several studies have shown that the genomes of more than 50% of tumors from selected human cancers show evidence of chromothripsis [140,141,142]. Similar to translocations seen in human cells, chromothripsis repair events are often mediated by cNHEJ and to a lesser degree by alt-EJ and MM-BIR [140,141,142].

### 4.7. Repair Pathway Choice in the Context of Genome Editing

The use of CRISPR-Cas systems has revolutionized molecular biology and provided a diverse tool kit for genome editing, therapeutics, and diagnostics. To fully realize the potential of this technology, a complete understanding of how Cas9-induced DSBs are repaired is needed. Specifically, the ability to predict CRISPR outcomes, including the relative balance of cNHEJ and alt-EJ, in genome editing experiments would be ideal. A study performed in mammalian cells with DSBs induced by SpCas9 and more than 40,000 different single guide RNAs (sgRNAs) illustrated that editing outcomes mostly depend on the targeted sequence itself. Strikingly, 11% of all sgRNAs tested had a dominant outcome that composed > 40% of all repair events for that sgRNA [143]. Analysis of repair outcomes showed that 58% of events involved a deletion of at least 3 bp and half of those used microhomology, indicative of alt-EJ repair (Figure 4). Three percent of all repair events were insertions of ≥ 1 bp and 99% of these were an insertion of the PAM distal nucleotide. These events were likely due to the fill-in of Cas9-induced DSBs with single nucleotide 5′ overhangs. In another study of Cas9-induced breaks, 5–10% were repaired by TMEJ under normal physiological conditions [37]. Thus, it appears clear that the ability to predict alt-EJ repair products will be critical to reach the goal of understanding the potential outcomes of Cas9 genome editing experiments.

The type of DNA ends can also impact the outcome of genome editing experiments (Figure 4). In mammalian cells, the use of Cas9 nickase variants can produce DSBs with either 3′ or 5′ overhangs. In one study, repair of DSBs with 3′ overhangs frequently produced insertions apparently templated from sequences adjacent to the break. These insertions were longer than those formed with 5′ overhangs, occurred more frequently, and use microhomologies indicative of TMEJ [144]. In contrast, DSBs with 5′ overhangs promoted HR more efficiently than 3′ overhangs and typically led to larger deletions. These deletions likely arose from the processing of the 5′ overhang into a 3′ overhang prior to repair by HR or alt-EJ [144].

## 5. Conclusions and Remaining Questions

Genomes are constantly assaulted by agents that cause DNA double-strand breaks. To deal with this, organisms rely on both end-joining and recombination-based repair pathways. These are highly regulated and usually result in accurate restoration of the genome. However, it is becoming clear that despite its mutagenic potential, error-prone DSB repair occurs with surprising frequency. Importantly, error-prone DSB repair is not simply a backup mechanism utilized when the accurate repair is not possible. Multiple factors influence its usage including enzymatic processes related to cell cycle stage, genome structure, and local sequence context. Error-prone repair is also purposeful—in its absence, a more serious genome catastrophe can occur.

A survey of the current literature demonstrates that error-prone break repair serves as a driving force of genome evolution, both in the germline and in somatic tissues in the context of cancer. It is increasingly apparent that a detailed understanding of how alt-EJ and other mutagenic repair mechanisms are regulated will have important therapeutic implications, in the contexts of disease treatment, aging, and genome editing.

To further this understanding, many questions remain to be answered. For example, how do local sequence variation and chromatin context affect alt-EJ repair, particularly TMEJ? What is the mechanism behind the rare but detrimental deletion-prone end joining that is observed in the absence of both cNHEJ and TMEJ? With what frequency does error-prone repair operate during mitosis and what are the consequences of this type of repair for genome stability? Why do different end-joining mechanisms preferentially mediate translocation formation in different organisms and how does this preference impact large scale genome evolution events like chromothripsis? Continued investigation of these questions and others will certainly provide additional surprises into the ways that error-prone repair is leveraged to drive genome evolution.

## Figures and Tables

**Figure 1 cells-09-01657-f001:**
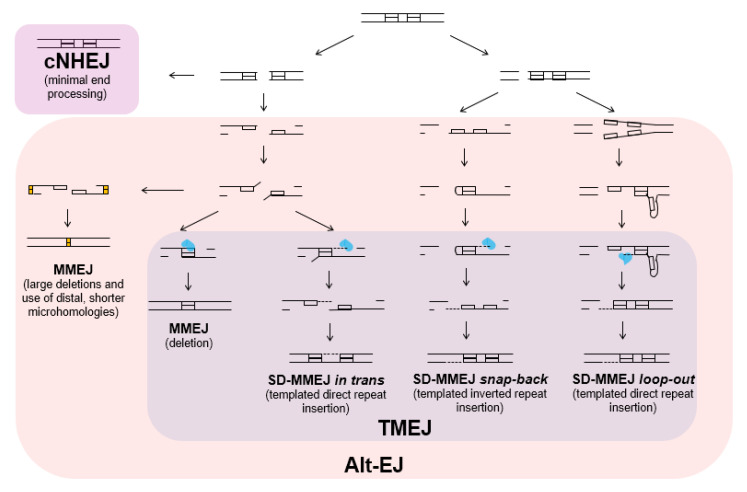
Categories of end-joining repair. The various forms of end joining are shown according to their mechanistic and genetic requirements. Alternative end joining (alt-EJ) is genetically distinct from canonical NHEJ (cNHEJ). Three models for synthesis-dependent microhomology-mediated end joining (SD-MMEJ) have been proposed, all of which can utilize DNA polymerase theta. In addition, simple MMEJ in metazoans (but not in yeast) requires polymerase theta. These types of MMEJ are grouped together as theta-mediated end joining (TMEJ). In the absence of polymerase theta, breaks can be repaired by a genetically undefined form of MMEJ that results in extremely large deletions.

**Figure 2 cells-09-01657-f002:**
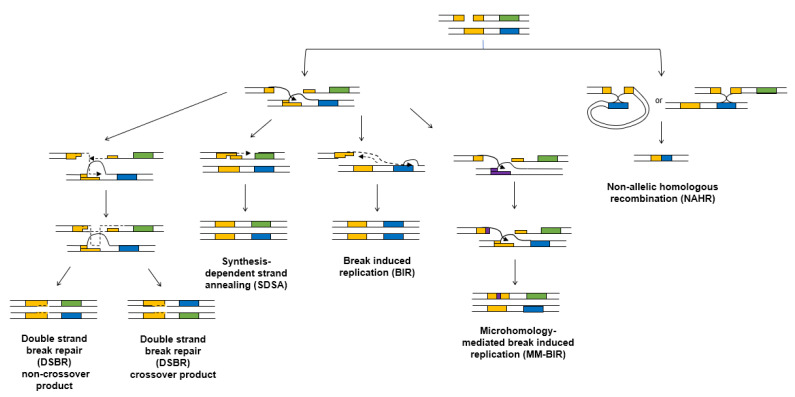
Mutagenic outcomes of homologous recombination. All types of homologous recombination begin with resection to form single-stranded DNA, followed by RAD51-mediated strand invasion. SDSA and DSBR usually use the sister chromatid as a template but the use of the homologous chromosome can result in loss of heterozygosity (SDSA) or mitotic crossovers (DSBR). BIR involves the formation of a mobile D-loop and produces long, single-stranded DNA tracts. BIR can result in loss of heterozygosity and mutations caused by damage in persistent ssDNA. MM-BIR involves multiple rounds of strand invasion, which can cause insertions of heterologous sequences. NAHR occurs when single-stranded DNA invades into a non-allelic template. NAHR can be intrachromosomal or interchromosomal and results in deletions (shown), duplications, and inversions.

**Figure 3 cells-09-01657-f003:**
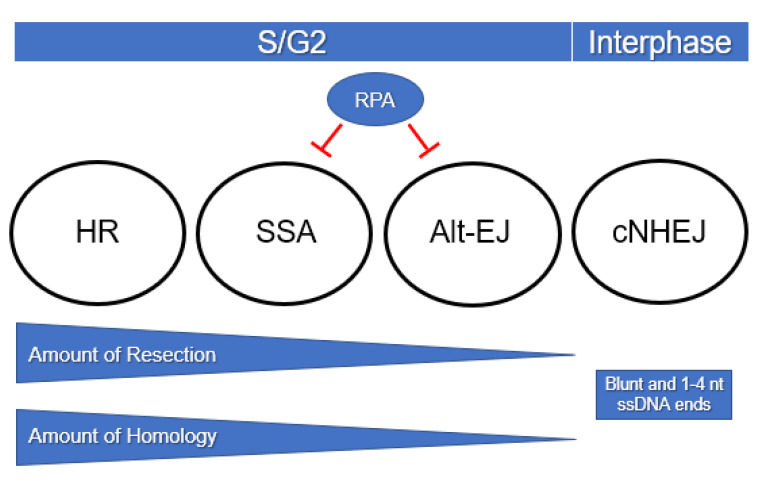
Factors that influence DSB repair pathway choice. cNHEJ operates throughout the cell cycle. HR, SSA, and alt-EJ require DNA resection and are therefore utilized primarily in S/G2. The required amounts of resection and homology are highest for HR and decrease for SSA and alt-EJ. RPA inhibits SSA and alt-EJ by preventing annealing of complementary sequences in single-stranded DNA.

**Figure 4 cells-09-01657-f004:**
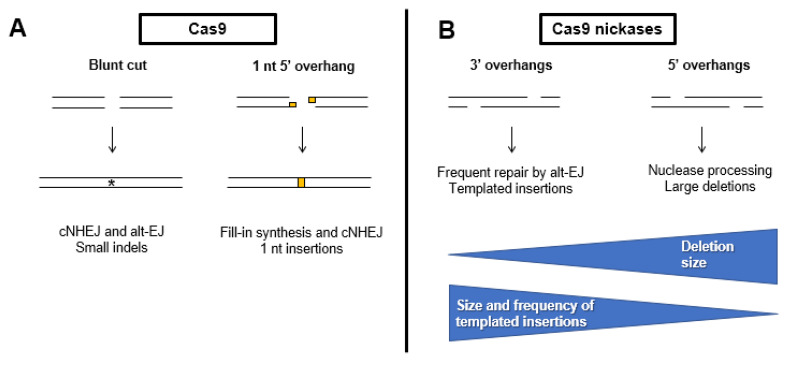
Mutagenic outcomes of Cas9-mediated genome editing. (**A**) Cas9-mediated double-strand breaks are usually blunt-ended and can be repaired inaccurately by cNHEJ and alt-EJ. Occasionally, Cas9 leaves 5′ single-stranded overhangs that are filled in, resulting in a single nucleotide insertion. (**B**) 3′ overhangs produced by Cas9 nickases are often repaired by alt-EJ, creating large, templated insertions. In contrast, 5′ overhangs are processed by nucleases to produce substrates suitable for alt-EJ and HR, sometimes resulting in large deletions.

**Table 1 cells-09-01657-t001:** Cancers reliant upon error-prone DSB repair. For each type of cancer/disease, the repair pathway that is impaired, genetic background, and the types of genomic mutations observed are listed. The druggable targets column indicates therapeutic approaches that may be used to exploit a synthetic lethal interaction. TD = tandem duplications, LOH = loss of heterozygosity.

Cancer/Disease	Affected Repair Pathway	Genetic Background	Drugable Targets	Effects on Genome	References
Breast, ovarian, melanoma, prostate, pancreatic	HR	BRCA1 deficient	POLQ, PARP1	Translocations, TDs, LOH, point mutations	[117]
Breast, ovarian	HR	BRCA2 deficient	POLQ, PARP1	Translocations, LOH, TDs, point mutations	[41]
Epithelial ovarian cancers	HR	FANCD2 deficient, Increased expression of POLQ	PARP1, FANCD2, POLQ	Chromosomal aberrations, nonsynonymous mutations	[118]
Breast, ovarian, Fanconi anemia	HR	PALB2 deficient	PARP1	Translocations, LOH, TDs, point mutations	[117]
Breast, stomach, prostate	HR	CHD1 deficient, Deficient in CtIP recruitment to DSBs	PARP1, PTEN	Translocations, LOH, TDs, point mutations	[119]
Breast, ovarian, Fanconi anemia	HR	RAD51C deficient	PARP1	Genomic instability, aneuploidy, chromosome aberrations	[120]
Breast, ovarian, Fanconi anemia	HR	RAD51D deficient	PARP1	Large deletions, genomic instability	[121]
Chronic myeloid leukemia	HR	BCR-ABL (constitutively active tyrosine kinase), increased expression of LIG3a, PARP1, and WRN	PARP1 combined with DNA ligase inhibitors	Translocations, LOH, TDs, point mutations	[122,123]
Lynch Syndrome, colorectal, endometrial, ovarian, gastric, urinary, small bowel, pancreatic, prostate	MMR, HR	MLH1, MSH2, MSH6, PMS2 deficient	PARP1	Point mutations, microsatellite instability	[124]
High grade bladder tumors, colon	cNHEJ	KU, DNA-PK, or XRCC4 deficient	Unknown	Deletions (<125 bp on average) and microhomology at repair junctions	[125]
Multiple myeloma, leukemia, pro B-cell lymphoma	cNHEJ	KU and P53 deficient	Unknown	Rearrangements and gene amplifications, nonreciprocal translocations, increased microhomologies at repair junctions	[126,127,128]
MCF7 breast cancer	cNHEJ	Reduced levels of DNA LIG4 Increased levels of DNA LIG3a and PARP1	PARP1 combined with DNA ligase inhibitors	Large deletions, translocations	[129]
Acute myeloid leukemia	cNHEJ	Decreased expression of KU proteins Increased expression of DNA LIG3a	Unknown	TDs, microhomology at repair junctions, deletions	[130]
Neuroblastoma	cNHEJ	LIG4 and Artemis deficient Increased expression of DNA LIG3, LIG1, and PARP1	PARP1	Translocations, TDs	[131]
Non-BRCA1/2 breast cancer	cNHEJ	XRCC4 deficient	Unknown	Translocations	[132]

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
