# Peer review of "Regulation of Error-Prone DNA Double-Strand Break Repair and Its Impact on Genome Evolution"

_cells, 2020, doi:10.3390/cells9071657_

Round 1
Reviewer 1 Report
I congratulate the authors with and excellent review. In my opinion, there are a few minor issues which they have to address in the revised manuscript.
- I would advise the authors to add an extra graph showing the homologous recombination repair pathway. It may be added to Figure 1.
- It would be very useful if the authors can add to the text the contribution of different pathways to the repair of radiation-induced double-strand beaks. On page 7 they mention the contribution of cNHEJ and HR. The question is what about other pathways?
Author Response
We appreciate your suggestions and have addressed both of them in our revision. Below, we describe our changes.
1. I would advise the authors to add an extra graph showing the homologous recombination repair pathway. It may be added to Figure 1.
We created a new figure (Figure 2) showing different models of homologous recombination, highlighting the ways in which they can be mutagenic (page 10).
2. It would be very useful if the authors can add to the text the contribution of different pathways to the repair of radiation-induced double-strand beaks. On page 7 they mention the contribution of cNHEJ and HR. The question is what about other pathways?
We added text describing the contributions of alt-EJ to repair of radiation-induced DSBs, particularly clustered breaks (page 13, lines 379-382).
Reviewer 2 Report
Review of Hanscom and McVey (ID: cells-859910)
The review by Hanscom and McVey on different classes of end joining and dsDNA break repair summarizes the history of discoveries in the field and reviews genetic and biochemical data to back up those discoveries. In addition, they discuss mechanisms and regulation of pathway choices and connections to evolution and to cancer/replication stress. The scope and depth of the review gives the reader a comprehensive viewpoint on the subject and is a very well written and balanced review. I support its publication with a few minor additions to make it even stronger.
1. Please add a Figure to graphically explain the importance of end joining mechanisms in genome editing technologies. Since this is such hot area, a Figure would help the reader understand its importance.
2. When I read a review, I always value a paragraph towards the end that summarizes the unresolved important questions that we don't understand and what tools are needed to address limitations in the field. Please add a paragraph to address important unknown areas or mechanisms and any technologies that are needed to move the field forward towards deeper understanding. This will complete the Review: a thorough background and context followed by future challenges and perhaps will inspire the reader to rise to these challenges.
Author Response
We appreciate your suggestions and have addressed both of them in our revision. Below, we describe our changes.
1. Please add a Figure to graphically explain the importance of end joining mechanisms in genome editing technologies. Since this is such hot area, a Figure would help the reader understand its importance.
We agree completely! We have now added Figure 4, which shows how cNHEJ, alt-EJ, and HR may contribute to error-prone repair during Cas9-mediated genome editing (page 27).
2. When I read a review, I always value a paragraph towards the end that summarizes the unresolved important questions that we don't understand and what tools are needed to address limitations in the field. Please add a paragraph to address important unknown areas or mechanisms and any technologies that are needed to move the field forward towards deeper understanding. This will complete the Review: a thorough background and context followed by future challenges and perhaps will inspire the reader to rise to these challenges.
We have added a paragraph at the end of the review highlighting unanswered questions related to error-prone DSB repair and genome evolution (page 28, lines 781-789).
Reviewer 3 Report
Authors have discussed details of various DSB repair mechanisms in the context of cell cycle, pathogenesis and evolution. The article focuses especially on systemizing the alternative-end-joining pathways, which is particularly valuable. The review is written in a comprehensive manner and reads well. Overall, the review provides a valuable synthesis of knowledge that otherwise is scattered over the literature. I recommend it for publication after the following minor changes:
- The positioning of Figure 1 precedes discussion of the mechanisms illustrated in it, it would be more convenient for the reader to have all the abbreviations used in the figure (e.g. MMEJ, SD-MMEJ etc.) developed in the figure legend.
- Figure 2 - correct the "RPA" label so that "A" is in line with "RP".
Author Response
We appreciate your suggestions and have made the requested changes in our revised manuscript.
1. The positioning of Figure 1 precedes discussion of the mechanisms illustrated in it, it would be more convenient for the reader to have all the abbreviations used in the figure (e.g. MMEJ, SD-MMEJ etc.) developed in the figure legend.
We have fully written out the names of the repair mechanisms corresponding to the abbreviations in Figure 1.
2. Figure 2 - correct the "RPA" label so that "A" is in line with "RP".
Thank you for catching this--it has been corrected.